# Four New Species of *Haplocauda*, with Notes on the Evolutionary Convergence of Copulation Clamps in Lucidotini (Coleoptera: Lampyridae: Lampyrinae)

**DOI:** 10.3390/insects16080824

**Published:** 2025-08-08

**Authors:** Leandro Zeballos, Luiz Felipe Lima da Silveira, Cláudio Ruy Vasconcelos da Fonseca

**Affiliations:** 1Laboratório de Sistemática e Ecologia de Coleoptera (LASEC), Coordenação de Biodiversidade (CBIO), Instituto Nacional de Pesquisas da Amazônia (INPA), Manaus 69067-375, AM, Brazil; claudioruy.fonseca@gmail.com; 2Biology Department, Western Carolina University, Cullowhee, NC 28723, USA; limadasilveiral@email.wcu.edu

**Keywords:** lucidotini, *Scissicauda*, firefly, Amazon, morphology

## Abstract

Recent collecting efforts combined with studies of specimens deposited in different scientific collections allowed us to identify four new species of South American fireflies with unique abdominal traits. A study of their evolutionary relationships based on morphological traits confirms they are most closely related to *Haplocauda*, fireflies from the Amazon Rainforest [in Brazil]. We described these new species and provided an identification key to species in this genus.

## 1. Introduction

Lucidotini Lacordaire, 1857 is the largest tribe of Lampyridae Rafinesque, 1815, with 34 genera and 849 species [1]. The scarcity of taxonomic studies and overlap of diagnostic characteristics among genera render their identification challenging. The taxonomy of these genera is still based on few, highly homoplastic morphological characteristics, lacking phylogenetic scrutiny [2,3]. The overlap among genus-level diagnoses is even more evident among the Lucidotina Lacordaire, 1857 and Photinina LeConte, 1881, which contain almost ¾ of the tribe’s diversity, due to the broad definitions of their namesakes with other Lucidotini genera. These challenges make species-level identification nearly impossible and call for revisionary works to improve the state of this group’s taxonomy.

Recently, the inclusion of traits of the terminalia and genitalia in diagnoses and phylogenetic analyses has been an important factor in revisiting the generic limits within Lucidotini. Exploring such traits in a phylogenetic context has improved stability in Lucidotini, but the latter is still challenged by a massive underestimation of their species richness, as descriptions of new species continue to amount across the Neotropics [4,5,6,7,8,9,10,11,12].

The poor state of Lucidotini taxonomy is reflected in the fact that some genera of Lucidotini do not fit into any of the existing subtribes. Recent phylogenetic analyses revealed a distinctive clade within Lucidotini, dubbed as the “*Scissicauda* lineage” (*Haplocauda* Silveira, Lima, and McHugh, 2022, *Scissicauda* McDermott, 1964, *Pyractonema* Solier, 1849 and *Pyropyga* Motschulsky, 1852). Among the synapomorphic characteristics of this lineage are the well-sclerotized and anteriorly notched submentum, the dorsal plate of the phallus broadly fused to the parameres, and the syntergite medially split, divided into two plates [5]. In this lineage, *Scissicauda* and *Haplocauda* are the best-studied morphologically and stand out for the interspecific diversity of the terminal abdominal segments, which are hypothesized to play a role during copulation and fertilization [5,6,7,13].

Although important for the taxonomy of *Scissicauda* and *Haplocauda*, these abdominal modifications have not yet been observed during copulation, and their mechanism remains unknown. In the meantime, these structures were found to bear great phylogenetic information [5,6]. One important open question is whether the increased-length abdominal segment VIII (in relation to segment VII) of the closely related genera *Scissicauda* and *Haplocauda* is synapomorphic or homoplastic. This question is important because it could help answer whether similar modifications in copulation-related morphology could arise multiple times, possibly in response to similar selective pressures.

Recent research on specimens collected during the “Rede temática: Biodiversidade de Insetos na Amazônia (Rede BIA)” project has identified four undescribed species of Amazonian fireflies that exhibit unique modifications like the hypothesized copulation clamp (sternum VIII and pygidium) found in males of *Haplocauda*. To evaluate the placement of these new species in *Haplocauda* and investigate their relationships with other Lucidotini taxa, we ran phylogenetic analyses based on Maximum Parsimony. Here, we describe and illustrate these four new species and provide identification keys based on males and maps showing the distribution of the species.

## 2. Materials and Methods

### 2.1. Morphology, Terminology and Map 

The type materials were located at the following institutions: Instituto Nacional de Pesquisas da Amazônia (INPA); Coleção Zoológica da Universidade Federal de Mato Grosso (CEMT); Coleção Entomológica Professor José Alfredo Pinheiro Dutra, Universidade Federal do Rio de Janeiro, Rio de Janeiro, Brazil, (DZRJ); Museu de Zoologia da Universidade de São Paulo, São Paulo, Brazil (MZUSP); Museu Paraense Emílio Goeldi (MPEG); and the Coleção de Entomologia Pe. Jesus Santiago Moure, Universidade Federal do Paraná, Curitiba, Brazil (DZUP).

Head, thorax (excluding hindwings and elytra) and abdomen were taken from 1–2 male and abdomen from 1–2 female specimens per species and soaked in 10% KOH for 24–36 h. Females were tentatively associated based on capture together with the male in the same collecting event and by overall similarity to the male. Specimen photographs were taken using a Leica DFC295 camera attached to a Leica M205C stereomicroscope with the Leica Application Suite X Automontage Software V3.4.1 (Version 2009) and Leica DFC295 camera attached to a Leica M165C stereomicroscope with the Leica Application Suite V3.4.1 Auto montage Software. We follow the classification of The Lampyridae of the World Database [1], the anatomical terminology of [7,14] for hind wings.

We built an occurrence map in R for the focal species of this work based on specimen collection labels. We colored the map by Neotropical dominions sensu [15], based on the shapefile given in [16], using the color palette in [6]. The following packages were used for the generation of the map: “sf” [17], “dplyr” [18], and “ggplot2” [19].

Label data for all type specimens were transcribed using the following conventions: double forward slashes (//) to separate labels; double comma („) for line breaks, and brackets [ ] to enclose our comments or notes. The map and plates were assembled using Adobe Photoshop 2021.

### 2.2. Phylogenetic Analysis

To support the classification of the four new species, we include them in a matrix with taxa from all four Lucidotini subtribes (total = 30 species), rooting in *Lampyris noctiluca* (Linnaeus, 1758) (Lampyridae: Lampyrini). Examined outgroup material used in the phylogenetic analyses is provided in Appendix A.

Our matrix analyzed 88 morphological characters listed in [5,13], with two revised characters (C16 and C91) and the addition of new characters (C30, C31, C71, 72; total = 92) to accommodate new states. Character coding followed [20]. The characters were arranged in Mesquite v3.81 [21]. The revised character matrix is provided as a nexus file in Appendix A.

We ran phylogenetic analyses on TNT v.1.5 [21], using new technology heuristic searches with TBR (with Ratched, Drift and Tree fusing active), with equal (EW) and implied weights (IW). For the IW analysis, the concavity constant (*k*) was calculated using the TNT script “setk.run”, written by Salvador Arias (Instituto Miguel Lillo). This script searches for the value of this constant (k) that presents the greatest congruence between the replications, according to the dataset entered in TNT, and its use reduces the bias of manually choosing a K value (K = 4.765625, in this study). Node support was assessed using the absolute Bremer index and Bootstrap values for MPEW and symmetric resampling for MPIW [22,23]. Strict consensus of the four most parsimonious trees found in EW and the two in IW as well as optimization of ambiguous characters and analysis of the evolution of the characters were made using Winclada 2.0 [24]. The final trees were edited using Adobe Photoshop 2021.

## 3. Results

This detailed study of the morphology of the taxa included in the analyses allowed us to identify 92 characters (males) (Appendix A): 8 from the head; 16 from the thorax; and 68 from the abdomen, 38 of which were from the aedeagus. The characters were coded as binary (N = 55) or multistate (N = 37). For each character, the following is indicated regarding the IW topology: the number of steps (L); the consistency index (CI); and the retention index (RI).

### 3.1. Phylogeny

The Maximum Parsimony (MP) analysis of the data matrix (Appendix A) under IW resulted in two trees, from which we generated a strict consensus tree (L = 306, CI = 45, RI = 73) (Figure 1). In contrast, MP under EW resulted in the four most parsimonious trees, from which we generated a strict consensus tree (L = 336, CI = 41, RI = 68) (Figure 2). In all topologies, *Haplocauda* species was recovered as a monophyletic group with high support in IW (symmetric resampling (93)) and low support in EW (absolute Bremer (3) and bootstrap (80) indices). The synapomorphies supporting *Haplocauda* in all trees are the following (Figure 1): Sternum VIII, length relative to VII: slightly longer (28:1, homoplastic) (Figure 3B); Sternum IX, lateral rods, shape: abruptly convergent (48:2, not homoplastic); Sternum IX, lateral rods, anterior thickening: present (52:1, not homoplastic); Sternum IX, posterior margin, shape: emarginated (54:2, not homoplastic); Phallus, dorsal plate, median fusion to parameres, extent: up to 1/3 of dorsal plate length (61:0, homoplastic); Phallus, dorsal plate, apical arms (of indented Phallus), shape: contiguous (70:1, not homoplastic); Phallus, ventral plate, shape: I-shaped (76:0, homoplastic).

Within the *Haplocauda* clade, EW and IW (Figure 1 and Figure 2) recovered very similar relationships with the exception of two species. *H. lata* **sp. nov.** was recovered as a sister to all other species, followed by *H. antimary* **sp. nov.** (supported by symmetric resampling (88), absolute Bremer (3) and bootstrap (69) indices), as a sister to the remainder species. While EW recovers the *H. amazonensis*
**sp. nov.** + *H. aculeata* **sp. nov.** branch sister to the clade containing the three previously described *Haplocauda* species, IW recovers a polytomy between *H. amazonensis*
**sp. nov.**, *H. aculeata* **sp. nov.** and the clade containing *H. yasuni* + (*H. mendesi* + *H. albertinoi*) (Figure 1 and Figure 2). The sister group of *Haplocauda* was *Scissicauda*, strongly supported by symmetric resampling (89), moderated by absolute Bremer (4) and Bootstrap (81). As in previous analyses (6), the “*Scissicauda* lineage” is consistently recovered in all trees, supported by absolute Bremer (1), Bootstrap (46) and symmetric resampling (85).

The relationships of the taxa neighboring the *Haplocauda* clade remained consistent across the topologies of both EW and IW, with only one polytomy recovered in the consensus tree (Figure 2). In addition to the “*Scissicauda* lineage” clade, both topologies recovered the branch *Dadophora hyalina* Duponchel, 1844 as sister to *Costalampys* spp. (supported by absolute Bremer (1), Bootstrap (44) and symmetric resampling (60)). However, while the other genera (*Uanauna angaporam* Campello-Gonçalves, Souto, Mermudes & Silveira, 2019, *Photinus* spp., *Alychnus suturalis* (Motschulsky, 1854), *Pseudolychnuris vittata* Motschulsky, 1853, *Lucidota banoni* Laporte, 1833, *Ybytyramoan monteirorum* Silveira & Mermudes, 2014) are recovered in a polytomy in EW (Figure 2), IW resolves this polytomy by placing (I) *P. vitatta* sister to the other Lucidotini (except *Ethra* Laporte, 1833) and the remaining taxa in a clade next to “*Scissicauda* lineage” branch, containing (II) *U. angaporam* at the base of this clade, (III) *Photinus* spp. as sister to *A. suturalis* and (IV) *L. banoni* next to *Y. monteirorum* (Figure 1).

### 3.2. Taxonomy

Lampyridae Rafinesque, 1815

Lampyrinae Rafinesque, 1815

Photinini Olivier, 1907

*Haplocauda* Silveira, Lima, and McHugh, 2022

**Type species:** *Haplocauda albertinoi* Silveira, Lima, and McHugh, 2022, by original designation.

**Diagnosis.** Antennae serrate, without branches (Figure 4G,N,U,Bb), tibial spurs 1-2-2 (1-2-1 in *H. yasuni* Silveira, Lima, and McHugh, 2022 and *H. amazonensis* **sp. nov.**). **Color pattern** (Figure 3): body overall brown, except for light-brown pronotal lateral expansions, and a longitudinal light-brown lateral stripe on elytron (outlined in brown or light brown); antennomeres IX–XI sometimes creamy white; legs light brown, sometimes with brown tibia and tarsus; pygidium sometimes with translucent lateral spots; sternum VIII with translucent lateral spots. **Male:** anterior claw of pro- and mesolegs with basal teeth; pygidium with anterior margin slightly or strongly emarginated, with apically blunt anterior projections (absent in *H. lata* **sp. nov.** and *H. antimary* **sp. nov.**), posterior margin without a median indentation, posterior angles acute or rounded; sternum VIII at least 2× longer than VII, with a posterior projection of variable length and width (absent in *Scissicauda*); sternum IX with lateral rods basally fused and posteriorly thickened, posterior margin of sternum IX rounded (emarginated in *H. lata* **sp. nov.**) with an acute projection (absent in *H. lata* **sp. nov.**, *H. aculeata* **sp. nov.** and *H. antimary* **sp. nov.**). **Female:** pygidium with posterior margin truncated or rounded; sternum VIII as long as wide and spiculum ventrale long and slender, as long as 3/4 or 2/3 sternum, with posterior margin slightly or deeply emarginate; internal genitalia with a large and somewhat rounded spermatophore-digesting gland and a lump-like spermatheca, bursa copulatrix with paired elongated and weakly sclerotized plates, with a very long accessory gland.


**Redescription**


**Head** entirely covered by pronotum when retracted (Figure 4A–F,H–M,O–T,V–Aa). Head capsule about 1.5–2× as wide as long, lateral margins slightly convergent posteriorly (Figure 4B); vertex somewhat convex. Antennal sockets reniform, slightly wider than distance between sockets, antennifer process conspicuous (Figure 4C). Antenna 11-segmented, scape constricted proximally, pedicel almost as long as wide and constricted basally, antennomeres III–X variably serrated and without lamellae, without upright bristles, antennomeres IX– or X–XI brown or creamy white, apical antennomere about as long as the subapical (Figure 4G,N,U,Bb). Frontoclypeus very wide, slightly curved (Figure 4C). Labrum connected to frontoclypeus by a membranous suture (Figure 4C); nearly 4× as wide as long, anterior margin evanescent (Figure 4A,C). Mandibles largely overlapping, long and slender, evenly arcuate, apex acute, without internal tooth, external margin sparsely setose in basal 1/2 (Figure 4A,C). Maxilla weakly sclerotized (Figure 4I); stipe about 2× as long as wide, posterior margins truncated, palpi 4-segmented, palpomere III subtriangular, IV lanceolate, with internal margin covered with minute, dense bristles, almost 2–3× longer than III (Figure 4C). Labium weakly sclerotized (Figure 4B); mentum completely divided sagittally, submentum sclerotized and bearing bristles, subcordiform or subrectangular, elongated, palpi 3-segmented, palpomere III securiform (Figure 4B). Gular sutures almost indistinct, gular bar transverse, nearly 3× as as wide as long (Figure 4B). Occiput piriform, 1/3 narrower than head capsule (Figure 4F).

**Thorax** with pronotum semilunar, with posterior angle rounded, disc subquadrate in dorsal view, almost flat in lateral view (Figure 5C,H,M,T), regularly punctured, punctures small and pubescent, with a line of distinct deep marginal punctures (Figure 5A,F,K,P); pronotal expansions well developed, anterior expansion maximal length almost 1/2 as long as disc, posterior expansions almost straight (Figure 5B); slightly wider than distance between elytral humeri (Figure 3). Hypomeron short in lateral view (width of lateral expansion of pronotum at least 1.5–2× greater than hypomeron depth) (Figure 5E). Prosternum 4 as wide as its major length, slightly narrowed parasagittally (Figure 5C). Proendosternite elongated, about as long as distance between the apices of proendosternite arms (Figure 5B,G,L,Q). Mesoscutellum with posterior margin rounded (Figure 3). Elytron ellipsoid, almost 2.6–3.5× as long as wide, pubescent, with a row of conspicuous punctures surrounding sutural and lateral margins (Figure 6A–D). Hind wing well developed, posterior margin slightly sinuate, 2× as long as wide; CuA2 present, mp-cu crossvein present; RP + MP1 + 2 3/4 r4 length, almost reaching distal margin, J indistinct (Figure 6E–H). Profemur about as long as protibia, meso and metafemora slightly shorter than respective tibiae. Tibial spur formula: protibia 1 (only posterior), mesotibia 2, metatibia 1 (only anterior) or 2. Male with anterior claw of pro and mesothoracic legs with basal tooth. Tarsomere I 2× longer than II, II 2× longer than III, III subequal in length to IV, IV bilobed, lobes reaching 2/3 V length. Metendosternum spatulate, 2 as long as wide, median projection acute anteriad, with two lateral laminae.

**Abdomen** with tergum I with anterior margin membranous, laterotergite membranous, nearly rounded, with sparse bristles; spiracle obliquely oriented on the thorax. Terga II–VII with posterolateral angles more produced and acute posteriorly, posterior margins more bisinuate. Sterna II–VIII visible. Spiracles dorsal, at about 1/2 sterna lengths.

**Male.** Sternum VIII 2× longer than VII, with a long and wide posterior projection (Figure 7A,G,P,W), with “larval” lanterns elongated. Pygidium with anterior margin strongly emarginated, with thick, sinuate, and apically blunt anterior projections; posterior margin without a median indentation; posterolateral angles acute (Figure 7B,J,Q,X). Syntergite medially divided, bearing bristles posteriorly, not anteriorly fused to sternum IX, 1/3 shorter than sternum IX, connate to sternum IX along its length (Figure 7C,K,R,Y). Sternum IX with lateral rods basally fused and posteriorly thickened, posterior margin of sternum IX with an acute projection (Figure 7K). Aedeagus with phallobase bilaterally symmetrical, sides abruptly convergent basally; dorsal plate of phallus basally fused to parameres, apically indented and with arms apically curved and approximate; parameres sinuate, nearly 1/5 longer than dorsal plate of phallus, lacking ventral rods; ventral plate rudimentary, restricted to a sclerotized piece outlining the opening of the ejaculatory duct or I-shaped, as long as 1/2 dorsal plate of phallus (in *H. amazonensis* **sp. nov.** and *H. antimary*
**sp. nov.**) (Figure 7D–F,M–O,T–V,Aa–Cc).

**Female.** Overall similar to male (Figure 8), except for the following traits. Metatibia with two spurs. Pygidium with posterior margin truncate (Figure 9E) or rounded (Figure 9H). Sternum VIII as long as wide, spiculum ventrale long and slender, 3/4 sternum length, with posterior margin moderately indented (Figure 9D) or deeply indented (Figure 9A,G). Internal genitalia with a large and somewhat rounded spermatophore-digesting gland and a lump-like (i.e., lacking a pedicel) spermatheca, bursa copulatrix with paired elongate, weakly sclerotized plates, or with eight irregularly shaped well-sclerotized plates [12] with a very long accessory gland (Figure 9J–L). Ovipositor (Figure 9C,F,I) with paraproctal valve 3× longer than paraproct core; coxites convergent posteriorly, well developed but weakly sclerotized; styli minute, sclerotized; proctiger plate entire, elongate, weakly sclerotized.

**Distribution.** Amazon Rainforest, South America (Figure 10).

**Remarks.** *Haplocauda* was classified in Photinini (Lampyridae: Lampyrinae) in the original description. Many taxonomic works of South American fireflies considered Photinini as valid, only recently synonymized with Lucidotini ([25] but made widely known via [1]). Among the Lucidotini included in our matrix, *Haplocauda* was found closely related to other genera of Lampyrinae: *Pyractonema* Solier, 1849 (Lucidotina), *Pyropyga* Motschulsky, 1853 (Photinina) and *Scissicauda* McDermott, 1964 (Photinina). This group has been called the “*Scissicauda* lineage” and might warrant a higher taxonomic rank recognition, like subtribe, in the future.

*Haplocauda* can be distinguished from *Scissicauda*, *Pyractonema* and *Pyropyga* by the median projection of sternum VIII, syntergite asymmetrical in length (except in *H. lata* **sp. nov.**), sternum IX with a pointed projection on the posterior margin. These characteristics, along with the presence of a tooth on the anterior tarsal claw of the pro- and mesoleg (present in all *Haplocauda* and *Scissicauda*) and the pygidium with anterior blunt projections (except in *H. lata* sp. nov. and *H. antimary* sp. nov., see discussion below), are also useful to differentiate *Haplocauda* from *Lucidota*, a genus with superficially similar species (like *Lucidota mellicula* Olivier 1907) which lacks a clear diagnosis and is taxonomically poorly delimited.

The median projection of the posterior margin of sternum VIII is the most diverse feature among *Haplocauda* species. In the original description of the genus, the posterior margin of sternum VIII with a long and wide posterior projection was sufficient to describe this structure, despite the clear differences in length and shape. However, here we describe species that present this narrow and extremely short (*H. lata* **sp. nov.**) or long and extremely narrow (*H. aculeata* **sp. nov.**) projection (Figure 7A,P).

The tentatively associated females of *Haplocauda* are morphologically similar to the females of *Scissicauda,* and the recent discovery of new species in these two genera has overlapped some characteristics that differentiated the females of these groups, such as the shape of the pygidium [5,13]. The females of *Haplocauda* can be distinguished from *Scissicauda* by having only one spur of protibia (two spurs in *Scissicauda*).

*Haplocauda lata* Zeballos and Silveira sp. nov.

(Figure 3, Figure 4, Figure 5, Figure 6 and Figure 7)

urn:lsid:zoobank.org:act:02C7915C-0F6F-4B07-8BAE-44EE14996F6E

**Diagnostic description.** Antennomeres entirely brown (Figure 4G). Pygidium brown with anterior corners mostly light brown to translucent (Figure 7B). Sternum VIII mostly light brown to translucent, with projection well sclerotized (Figure 7A). **Male.** Metatibia with two spurs. Pygidium with anterior blunt projections absent, central 1/3 indented, margin posterior bisinuate (Figure 7B). Sternum VIII with posterior projection short, as long as 1/5 the width of sternum VIII, basally as wide as 1/5 sternum width (Figure 7A). Sternum IX with margin posterior emarginate, acute projection absent (Figure 7C). **Female** (Figure 8A,B). Pygidium 1.5× as long as wide, with anterior margin emarginated, posterior margin truncated (Figure 9B). Sternum VIII with spiculum ventrale almost 1/3 shorter than sternum, posterior margin strongly indented (Figure 9A). Internal genitalia with spermatophore-digesting gland larger than spermatheca, bursa copulatrix with plate irregularly shaped, well sclerotized (Figure 9J). Ovipositor with valvifers free, twisted basally, 2× longer than coxite; coxites convergent posteriorly, divided in distinct proximal and distal plates, both well developed but weakly sclerotized; styli minute, sclerotized; proctiger plate weakly sclerotized (Figure 9C).


**Immature stages unknown.**


**Etymology.** The species epithet *lata* is a Latin adjective meaning “wide” or “broad.” It refers to the total width of the elytra, which is noticeably greater in this species compared to its congeners. Adjectival name.

**Type material.** HOLOTYPE (1♂, dissected, INPA), Brasil, Rondônia, Porto Velho,, ESEC Morro 3 Irmãos, vi.2017,, 9°00′09″ S—64°32′40″ W, Malaise,, V. S. Silva & J. A. Rafael- Rede BIA// Holotype Haplocauda lata [tracing paper, handwritten]. PARATYPES (27♂ and 10♀). (3♂, one dissected, alcohol, INPA), Brasil, AM, Tapauã,, Rio Ipixuna, Porto Coutinho,, 13–18.x.2013, Terra- Firme,, Malaise, D. M. M. Mendes Leg// Paratype Haplocauda lata [tracing paper, handwritten]. (2♂, alcohol, INPA),Brasil, AM, Careiro Castanho,, Br 319 Km 181, Sítio S. Paulo,, 4°12′48″ S—60°49′04″ W,, 16–31.xii.2016. Malaise G.,, J.A. Rafael & F. F. Xavier F°// Paratype Haplocauda lata [tracing paper, handwritten]. (1♀, alcohol), same, except for the date, 14–27.iii.2011// Paratype Haplocauda lata [tracing paper, handwritten]. (2♂, alcohol), same, except for the date, 16–31.i.2017// Paratype Haplocauda lata [tracing paper, handwritten]. (1♂, 1♀, alcohol), BR-AM, Tapauã, Rio,, Ipixuna, Porto Coutinho,, 08–13.x.2013. Terra Firme,, Malaise. D. M. M. Mendes,, leg.// INPA-COL,, 001572// Paratype Haplocauda lata [tracing paper, handwritten]. (1♂ and 1♀, alcohol, INPA), BR-AM, Ipixuna, Porto,, Coutinho,, 13–18.x.2013,, Terra Firme. Malaise,, D. M. M. Mendes leg.// INPA-COL,, 001648// Paratype Haplocauda lata [tracing paper, handwritten] (3♂, 1♀ pinned, UFMT), Brasil, Amazonas, Careiro,, Castanho, PPBIO, 03°40′42″,, S, 60°19′41″ W, 11–13.xii.,, 2013, malaise, J.A.Rafael,, J.T.Camara & F.F.Xavier F°// Paratype Haplocauda lata [yellow label]. (1♂, pinned, MZUSP), BR. AM. Manaus ZF-03,, BR 174 Km 41 Res. 1401,, Gavião,, 02°24′09″ S/59°49′45″ W// 16–31/X/1995,, Rocha e Silva, L.E.F. col.// 0061942// Malaise// Paratype Haplocauda lata [yellow label]. (1♂, pinned, MZUSP), the same as above except for the following label 0061938// Paratype Haplocauda lata [yellow label]. (1♂, pinned, MZUSP), the same as above except for the following label 0061939// Paratype Haplocauda lata [yellow label]. (1♂, pinned, MZUSP), the same as above except for the following label 0061934// Paratype Haplocauda lata [yellow label]. (1♂, pinned, MZUSP), the same as above except for the following label 0061941// Paratype Haplocauda lata [yellow label]. (1♂, pinned, MZUSP), the same as above except for the following label 0061937// Paratype Haplocauda lata [yellow label]. (1♀, pinned, MZUSP), the same as above except for the following label 0061921// Suspensa// Paratype Haplocauda lata [yellow label]. (1♂, pinned, INPA), Brasil, AM, Resex Urini,, Rio Urini, Lago 3 bocas,, 01°34′56″ S, 62°58′28″ W,, 14–22.vii.2004// Luz mista de,, mercúrio + BLB (lençol),, A. Silva F. & L. Aquino// INPA-COL,, 001194// Paratype Haplocauda lata [yellow label]. (1♂, pinned, INPA), BRASIL, AM. Parque,, Nacional do Jaú, Rio Urini,, 01°40′31″ S-61°46′34″ W,, 20–23.xi.1996. Malaise,, A. Henriques & J. Vidal/ INPA-COL,, 001242// Paratype Haplocauda lata [yellow label]. (1♂, pinned, DZUP), BR-AM, Autaz Mirin,, 15–30.i.1995; Bindá,, Arm Malaise// INPA-COL,, 001357// Paratype Haplocauda lata [yellow label]. (1♂, pinned, DZUP), BR-AM, Autaz Mirin,, 01–15.i.1995.,, Bindá col.// INPA-COL,, 001358// Paratype Haplocauda lata [yellow label]. (1♂, pinned, DZUP), the same as above except for the following label INPA-COL,, 001356// Paratype Haplocauda lata [yellow label]. (1♂, pinned, DZUP), the same as above except for the following label INPA-CO,, 001359// Paratype Haplocauda lata [yellow label]. (1♂, pinned, DZUP), the same as above except for the following label INPA-COL,, 001360// Paratype Haplocauda lata [yellow label]. (1♂, pinned, DZUP), the same as above except for the following label INPA-COL,, 001361// Paratype Haplocauda lata [yellow label]. (1♂, pinned, DZUP), the same as above except for the following label INPA-COL,, 001362// Paratype Haplocauda lata [yellow label]. (1♀, pinned, INPA), BRASIL, AM, Resex Urini,, Rio Urini, Lago 03 Bocas,, 01°34′13″ S, 62°58′54″ W,, 14–28.vii.2004// Arm. Malaise terra firme,, M.L.Oliveira, A. Silva F.,, L. Aquino leg.// Paratype Haplocauda lata [yellow label]. (2♀, pinned, INPA), BRASIL-PA-Melgaço,, Caxiuanã-ECFPn,, 23.iii.1998,, O. Silveira, J. Pena col.// Coleoptera: Polyphaga,, Elateriformia: Cantharoidea:,, Lampyridae,, Incorp: 22/iii/2002// Paratype Haplocauda lata [yellow label]. (1♀, pinned, INPA), BRASIL-PA-Melgaço,, Caxiuanã-ECFPn,, 23.iii.1998,, O. Silveira, J. Pena col.// Arm. S-Malaise,, SME 1,, ECFPn,, 23.iii.1998// Coleoptera: Polyphaga,, Elateriformia: Cantharoidea:,, Lampyridae,, Incorp: 22/iii/2002// Paratype Haplocauda lata [yellow label]. (1♂, pinned, INPA), BRASIL-PA-Melgaço,, Caxiuanã-ECFPn,, 23.iii.1998,, O. Silveira, J. Pena col.// Arm. S-Malaise,, SME 4,, Igarapé Curuazinho,, 23–28.iii.1998// Coleoptera: Polyphaga,, Elateriformia: Cantharoidea:,, Lampyridae,, Incorp: 22/iii/2002// Paratype Haplocauda lata [yellow label]. (1♀, pinned, INPA), BRASIL-PA-Melgaço,, Caxiuanã-ECFPn,, 23.iii.1998,, O. Silveira, J. Pena col.// Arm. S-Malaise,, SME 5,, ECFPn,, 20–29.iii.1998// Coleoptera: Polyphaga,, Elateriformia: Cantharoidea:,, Lampyridae,, Incorp: 22/iii/2002// Paratype Haplocauda lata [yellow label].

**Distribution.** Brazil: Amazonas: Careiro Castanho, Manaus, São Gabriel da Cachoeira, Tapauá, Resex Rio Unini; Pará: Melgaço; Rondônia: Porto Velho (Type locality) (Figure 10).

**Remarks.** *Haplocauda* lata **sp. nov.** is one of the largest species in the genus, along with *H. albertinoi* Silveira, Lima, and McHugh, 2022, and the second species described with the antenna entirely brown (the other one is *H. mendesi* Silveira, Lima, and McHugh, 2022). This species is unique among its congenerics due to the extremely short median projection of sternum VIII, wide elytra, pygidium with anterior margin emarginate, with anterior blunt projections absent and sternum IX with an emarginate posterior margin (Figure 7B,C). The species studied in this work were collected by Malaise and light traps. It is the most widely distributed species to date, being found in three Brazilian states: Amazonas, Pará and Rondônia.

*Haplocauda amazonensis* Zeballos and Silveira sp. nov.

(Figure 3, Figure 4, Figure 5, Figure 6 and Figure 7)

urna:lsid:zoobank.org:act:02C7915C-0F6F-4B07–8BAE-44EE14996F6E

**Diagnostic description.** Antennomeres X–XI creamy white (Figure 4N). Pygidium brown to dark brown with anterior blunt projections mostly light brown to translucent (Figure 7J). Sternum VIII with anterior corners light brown to translucent, with two rounded brown parasagittal spots, with projection well sclerotized. **Male.** Metatibia with one spur. Pygidium with anterior blunt projections well developed, subparallel-sided and with inner margin almost straight, posterior margin with central 1/3 slightly emarginated (Figure 7J). Sternum VIII with posterior projection basally as wide as ¼ sternum width, posterior projection with two basal apophyses (Figure 7G–I, blue arrows), acute, almost reaching the posterior margin of the pygidium (Figure 7G). Sternum IX with acute projection present, 1/5 longer than syntergite (Figure 7K). **Female** (Figure 8C,D). Pygidium 1.3× as long as wide, with anterior margin emarginated, posterior margin almost straight (Figure 9E). Sternum VIII with spiculum ventrale almost 1/3 shorter than sternum, posterior margin strongly indented (Figure 9D). Internal genitalia with spermatophore-digesting gland larger than spermatheca, bursa copulatrix with plate irregularly shaped, well sclerotized (Figure 9K). Ovipositor with valvifers free, twisted basally, 2× longer than coxite; coxites convergent posteriorly, divided in distinct proximal and distal plates, both well developed but weakly sclerotized; styli minute, sclerotized; proctiger plate weakly sclerotized (Figure 9F).

**Etymology.** “Amazonensis” is a Portuguese gentilic meaning “born in Amazonas,” referring to the species’ origin in the Brazilian state of Amazonas, where Portuguese is the official language. Adjectival name.

**Type material.** HOLOTYPE (1♂, dissected, INPA), BR- AM, Manaus,, Reserva Ducke,, 14.ii—06.iii.2007,, Platô Norte/Sul,, Arm. Malaise,, G. Freitas & M. Feitosa cols// Holotype Haplocauda amazonensis [tracing paper, handwritten]. PARATYPES (16♂ and 14♀). (1♀, alcohol, INPA), BR- AM, Ipixuna Porto,, Coutinho,, 13–18.x.2013,, Terra Firme. Malaise,, D. M. M. Mendes leg.// Paratype Haplocauda amazonensis [tracing paper, handwritten]. (1♂, alcohol, INPA), BR- AM, Manaus, Res. Campina,, Br 174 Km 43, 27.iii-08.iv.2009,, Malaise. B. B. Souza; C. O. Kiest; G.,, Freitas; J. J. Mendes; R. Freitas,, Silva// Paratype Haplocauda amazonensis [tracing paper, handwritten]. (1♂, alcohol, INPA), Brasil- Am, Careiro Castanho,, Br 319 Km 181, Sítio São,, Paulo 4°12′48″ S—60°49′04″ W,, 16–31.xii.2016. Malaise G.,, J. A. Rafael & F. F. Xavier F°// Paratype Haplocauda amazonensis [tracing paper, handwritten]. (1♂, alcohol, INPA), BR- AM, Manaus, Reserva,, Ducke,, Platô Leste/Oeste,, 16–30.xi.2006. Malaise,, Freitas; J. Vidal; G. Freitas cols.// Paratype Haplocauda amazonensis [tracing paper, handwritten]. (1♂, alcohol, INPA), BRASIL, AM, Manaus, Reserva,, Biológica de Cuieiras ZF-2,, S 02°35′21″ W60°06′55″, 06–09.viii,, 2013. Agudelo, A.; Maldaner, C.,, Malaise [the trap was handwritten]// Paratype Haplocauda amazonensis [tracing paper, handwritten]. (1♂, alcohol, INPA), BR- AM, Manaus, ZF2,, Km 14 2°35′21″ S–,, 60°06′55″ W, 17–31,, viii.2016. Malaise,, peq. Igarapé perto,, J. A. Rafael & F. F. Xavier F°// Paratype Haplocauda amazonensis [tracing paper, handwritten]. (1♂, alcohol), BR- AM, Manaus, Reserva,, Ducke. Platô Norte/Sul,, 09–20.x.2006. Malaise,, J. Vidal; R. Ale-Rocha; G. Freitas cols.// Paratype Haplocauda amazonensis [tracing paper, handwritten]. (4♂, alcohol), Brasil, AM, Manaus,, Reserva Ducke,, CMT1, v.1995// Paratype Haplocauda amazonensis [tracing paper, handwritten]. (1♂, pinned, UFMT), BRASIL, Amazonas, Careiro,, Castanho, PPBio, 03°40′42″,, S. 60°19′41″ W, 11–13. Xii,, 2013, malaise. J. A. Rafael,, J.T.Camara & F. F. Xavier F°.// Paratype Haplocauda amazonensis [tracing paper, handwritten]. (1♂, pinned, UFMT), BRASIL, Amazonas, Careiro,, Castanho, BR 319, Km-181,, Sítio S. Paulo, 4°12′48″ S,, 60°49′04″ W, 15–30. Vi,, 2016,, malaise. J. Rafael, F. Xavier F°.// Paratype Haplocauda amazonensis [tracing paper, handwritten]. (1♂, pinned, MZUSP), Brasil Amazonas,, Am. 010. Km. 25,, Reserva Ducke// Emergence trap// 9-xi-1977,, J. Arias// Paratype Haplocauda amazonensis [tracing paper, handwritten]. (1♂, 1♀, pinned, MZUSP), BR-AM, Manaus, Reserva,, Ducke 31.09.1996,, Ulisses Luiz// Paratype Haplocauda amazonensis [tracing paper, handwritten]. (1♂, 1♀, pinned, MZUSP), BRASIL, AM, Manaus,, Rod. Am 010 Km-26,, Res. Ducke ix.2001, J. F.,, Vidal, Malaise, mata.// Paratype Haplocauda amazonensis [tracing paper, handwritten]. (1♂, pinned, DZUP), BRASIL-AM, Manaus,, Reserva Florestal Ducke,, Igarapé Bolívia 28.ii.2003,, J.M.F. Ribeiro col.// Paratype Haplocauda amazonensis [tracing paper, handwritten]. (1♀, pinned, MZUSP), BRASIL, Amazonas,, Reserva F Ducke,, Igarapé Tinga. Malaise,, 05–16.viii.2004,, Henriques, A. leg// Paratype Haplocauda amazonensis [tracing paper, handwritten]. (1♀, pinned, DZUP), BRASIL-AM, Manaus,, Reserva Florestal Ducke,, Igarapé Ipiranga iv.2003,, J.M.F. Ribeiro col.// Paratype Haplocauda amazonensis [tracing paper, handwritten]. (1♀, pinned, DZUP), BRASIL-AM, Manaus,, Reserva Floprestal Ducke,, Igarapé Uberê xii.2002,, J.M.F. Ribeiro, Jailson Vidal & J. Vidal col.// Paratype Haplocauda amazonensis [tracing paper, handwritten]. (2♀, pinned, DZUP), BR-AM, Manaus, Reserva,, Ducke 16.09.1996,, Ulisses Luiz// Paratype Haplocauda amazonensis [tracing paper, handwritten]. (2♀, pinned, UFMT), BRASIL: Amazonas, Ma-,, naus, ZF2 Km14,, 02°35′21″ S, 60°06′55″ W, 16-,, 30.ix.2016, malaise igarapé,, J.A. Rafael & F. F. Xavier F°.// Paratype Haplocauda amazonensis [tracing paper, handwritten]. (1♀, pinned, UFMT), BRASIL: Amazonas, Careiro,, Castanho, BR319, Km181,, Sítio S. Paulo, 4°12′48″ S,, 60°49′04″ W, 05–17.x.2016,, malaise, J. Rafael, F. Xavier F°// Paratype Haplocauda amazonensis [tracing paper, handwritten]. (1♀, pinned, UFMT), BRASIL: Amazonas, Ma-,, naus, ZF2Km14 tower,, 02°35′21″ S, 60°06′55″ W,, 01–15.ii.2017, malaise,, J.A. Rafael & F. F. Xavier F°.// Paratype Haplocauda amazonensis [tracing paper, handwritten]. (1♀, pinned, INPA), BRASIL-AM, Manaus,, Reserva Florestal Ducke,, Igarapé Ipiranga v.2003,, Malaise. J.M.F. Ribeiro col Paratype Haplocauda amazonensis [tracing paper, handwritten]. (1♀, pinned, INPA), BRASIL: Am,, Reserva Ducke,, 26 Km NE Manaus,, Barbosa, M. G. V.// Plot A,, Malaise 5,, Mai. 1995// Paratype Haplocauda amazonensis [tracing paper, handwritten].

**Distribution.** Brazil: Amazonas: Manaus (Type locality) and Careiro Castanho (Figure 10).

**Remarks.*** Haplocauda amazonensis* **sp. nov.** is the only species in this genus that has two basal apophyses on the posterior projection of Sternum VIII (Figure 7G–I, blue arrows), anterior margin of the pygidium U-shaped and bearing blunt anterior projections (as in *H*. *mendesi* Silveira, Lima, and McHugh, 2022). The syntergite of this species is strongly asymmetric, as in *H*. *antimary* **sp**. **nov**. (Figure 7L). The specimens studied in this work were collected using Malaise traps.

*Haplocauda aculeata* Zeballos and Silveira sp. nov.

(Figure 3, Figure 4, Figure 5, Figure 6 and Figure 7)

urn:lsid:zoobank.org:act:EA44E79F-4C8D-4A65-A6EA-9136306F5B58

**Diagnostic description.** Antennomeres X–XI creamy white (Figure 4U). Pygidium brown with anterior corners mostly light brown to translucent (Figure 7Q). Sternum VIII with anterior corners light brown to translucent, with projection well sclerotized. **Male.** Metatibia with two spurs. Pygidium with anterior blunt projections present, posterior margin with central 1/3 almost straight (Figure 7Q). Sternum VIII with posterior projection basally as wide as 1/4 sternum width, acuminate, reaching the posterior margin of the pygidium (Figure 7P). Sternum IX with acute projection absent, almost 1/3 longer than syntergite (Figure 7R). **Female** (Figure 8E,F). Pygidium 1.6× as long as wide, with anterior margin emarginate, posterior margin rounded (Figure 9H). Sternum VIII with spiculum ventrale almost 1/3 shorter than sternum, posterior margin moderately indented (Figure 9G). Internal genitalia with spermatophore-digesting gland larger than spermatheca, bursa copulatrix without sclerotized plate (Figure 9L). Ovipositor with valvifers free, twisted basally, almost 3× longer than coxite; coxites convergent posteriorly, divided in distinct proximal and distal plates, both well developed but weakly sclerotized; styli minute, sclerotized; proctiger plate weakly sclerotized (Figure 9I).

**Etymology.** “Aculeata” is a Latin adjective meaning “sting-bearing” and refers to the prominent median projection of sternum VIII on the male abdomen. Adjectival name.

**Type material.** HOLOTYPE (1♂, alcohol, INPA), BR, AM, Reserva Ducke,, 22.xi- 8.xii. 2014,, Malaise. Silva Neto,, A. M. Mendes// Holotype Haplocauda aculeata [tracing paper, handwritten]. PARATYPES (23♂ and 8♀). (1♂, alcohol, INPA), Brasil- AM,, P. N. do Jaú,, Lago Miratucu,, 24–25.vii.93,, L. S. Aquino leg,, em Pensilvânia// Paratype Haplocauda aculeata [tracing paper, handwritten]. (7♂, 3♀, alcohol, INPA), BR, AM, Pq. N. do Jaú,, 29.vii- viii. 2001,, 0154275 S, 613510W,, Arm. Malaise,, Campinarana baixa,, Henriques & Vidal// Paratype Haplocauda aculeata [tracing paper, handwritten]. (1♂, alcohol), BRASIL, AM,, Pq. Nacional do,, Jaú 28.vii- viii.2001,, 01°54′27″ S; 61°35′10″ W,, Malaise. Campinarana,, Baixa Henriques & Vidal// Paratype Haplocauda aculeata [tracing paper, handwritten]. (1♂,1♀, alcohol), BR, AM, Pq. N. do Jaú,, 29.vii- viii. 2001,, 0154275 S, 613510W,, Arm. Malaise,, Campinarana baixa,, Henriques & Vidal// Paratype Haplocauda aculeata [tracing paper, handwritten]. (1♀, pinned, UFMT), BRASIL. EST. DO AMAZONAS,, Mun. São Gabriel da Cachoeira,, Querari 2° Pelotão de Fronteira,, (2° PEF) 01°05′N/ 69°51″ W// 05/IV- 27/V/ 1993,, Motta, C.S. Ferreira, R.L., Vidal J.,, & Matteo, B. col// Malaise// 0065672// Paratype Haplocauda aculeata [yellow label]. (1♀, pinned, MZUSP), BRASIL AM QUERARI,, São Gabriel da Cachoeira,, 2° Pel. Esp. De Fronteira,, 01°05′N/ 69°51″ W// 05/4- 27/V/ 1993,, Vidal, J; Ferreira, RLM col.// Malaise// 0065686// Paratype Haplocauda aculeata [yellow label]. (7♂, 1♀, pinned, MZUSP [3], DZUP[5]), BRASIL, Amazonas,, Parque Nacional do Jaú,, 29.vii—08.viii.2001,, 015446S, 613523W// Arm. Malaise,, Transição Campina,, Campinarana baixa,, Henriques & Vidal// Paratype Haplocauda aculeata [yellow label]. (1♂, 1♀, pinned, UFMT), BRASIL, Amazonas,, Parque Nacional do Jaú,, Campinarana Alta,, 01°53′42″ S; 61°35′10″ W,, Armadilha Malaise// 08 a 16.iv.2001,, Henriques & Vidal leg// Paratype Haplocauda aculeata [yellow label]. (1♂, pinned, UFMT), BRASIL AM QUERARI,, São Gabriel da Cachoeira,, 2° Pel. Esp. De Fronteira,, 01°05′N/ 69°51″ W// 05/4- 27/V/ 1993,, Vidal, J; Ferreira, RLM col.// Malaise// 0065704// Paratype Haplocauda aculeata [yellow label]. (1♂, pinned, INPA), BR-AM, Pq. Nacional do,, Jaú, Seringal. 27.iv-03.v.,, 1995. Arm. Malaise,, J. A. Rafael & J. Vidal// Paratype Haplocauda aculeata [yellow label]. (1♂, pinned, INPA), BR-AM, Parnajaú-Mata,, 19.vii-09.viii.2001. Malaise,, Igarapé. Terra Firme,, A. Henriques & Vidal// Paratype Haplocauda aculeata [yellow label]. (1♂, pinned, INPA), BRASIL, Amazonas, Pq. N.,, Jaú, Rio Carabinani,, Boa vista, 020105S-,, 613219W, 29–31.vii.1995// Arm. Malaise, J. A.,, Rafael & J. Vidal// Paratype Haplocauda aculeata [yellow label]. (1♂, pinned, INPA), BR, AM, Parque Nacional,, do Jaú, Rio Unini,, 20–24.vi.1996 Malaise,, A.L. Henriques, J. Vidal &,, F. L. Oliveira// Arm. Malaise, J. A.,, Rafael & J. Vidal// Paratype Haplocauda aculeata [yellow label].

**Distribution.** Brazil: Amazonas: Novo Airão (Type locality), São Gabriel da Cachoeira and Manaus (Figure 10).

**Remarks.*** Haplocauda aculeata* **sp. nov.** is the only species that has sternum VIII with a long and narrow posterior projection (short in *H*. *lata* **sp. nov.**) (characteristic that inspired the name of the species) and pygidium with anterior margin C-shaped, with anterior blunt projections present (absent in *H*. *antimary* **sp. nov.**) (Figure 7P,Q). The species studied in this work were collected using Malaise traps.

*Haplocauda antimary* Zeballos and Silveira sp. nov.

(Figure 3, Figure 4, Figure 5, Figure 6 and Figure 7)

urn:lsid:zoobank.org:act:5E8B859A-A590-40F3-947F-54816EA30E74

**Diagnostic description.** Antennomeres X–XI creamy white (Figure 4Bb). Pygidium brown with anterior corners mostly light brown to translucent (Figure 7Y). Sternum VIII with anterior corners light brown to translucent, with projection well sclerotized. **Male.** Metatibia with two spurs. Pygidium with anterior blunt projections absent, anterior margin emarginate, posterior margin with central 1/3 almost straight (Figure 7Y). Sternum VIII with posterior projection basally as wide as 1/5 sternum width, almost 3× longer than sternum VIII length, not reaching the posterior margin of the pygidium, apex rounded (Figure 7W). Sternum IX with posterior margin rounded, without acute projection, as long as aedeagus (Figure 7Z).


**Female and immature stages unknown.**


**Etymology.** The specific epithet refers to the Antimary State Forest, located in the State of Acre, where the type specimens were collected. Noun in apposition.

**Material Type.** HOLOTYPE (1♂, dissected, alcohol, INPA), BRASIL, Acre, Bujari, FES Antimary,, 9°20′01″ S- 68°19′17″ W, 3.viii-8.ix.,, 2016,, Malaise grande. E. F. Morato & J. A. Rafael cols—Rede BIA// Haplocauda antimary [tracing paper, handwritten]. PARATYPES (3♂). (1♂, dissected, alcohol, INPA), same as holotype. (2♂, alcohol, INPA), BRASIL, Acre,, 22–26.ix.2020,, pensilvânia UV// Paratype Haplocauda antimary [tracing paper, handwritten].

**Distribution.**
Brazil: Acre: Bujari (Figure 10).

**Remarks. ***Haplocauda antimary* sp. nov. is similar to *H*. *mendesi* Silveira, Lima, and McHugh, 2022 in that it presents sternum VIII with a short and rounded posterior projection. Unlike *H*. *mendesi*, H. *antimary* **sp. nov.** presents the pygidium with an anterior margin trisinuose, without anterior blunt projections (Figure 7Y). The specimens studied in this work were collected using Malaise and Pennsylvania traps.

*Haplocauda mendesi* Silveira, Lima, and McHugh, 2022

**New record.** (1♂, one Pinned, UFMT), Brasil: Mato Grosso, Alta,, Floresta, P. E. Cristalino, 9°,, 32′45″ S, 55°54′57″ W, FIT,, 15–19.ix.2019, R. A. Azevedo. Plot A FIT 4–2 (1500)// Halocauda mendesi.


**Key to *Haplocauda* Species Based on Males (Modified from Silveira et al., 2022)**


**1** Sternum VIII up to 1.5× wider than long (excluding the median projection of posterior margin); sternum IX with posterior margin rounded, with or without a projection (Figure 7).................................................................................................................................**2****1′** Sternum VIII almost 2× wider than long (excluding the median projection of posterior margin); sternum IX with posterior margin emarginate (Figure 7A–F).........................................................................................................................***H. lata* sp. nov.****2** Pygidium with anterior margin bearing blunt projections……………...........................**3****2′** Pygidium without anterior blunt projections on the anterior margin (Figure 7Y)................................................................................................................***H. antimary* sp. nov.****3** Sternum VIII with posterior projection acute (Figure 7A,G,P) ……………...……...…..**4****3′** Sternum VIII with posterior projection rounded (Figure 7W); ………………...…….…**6****4** Sternum IX with an acute projection (Figure 7K); metatibia with one spur …………**5****4′** Sternum IX without an acute projection (Figure 7R); metatibia with two spurs ….....................…...…………..........................……..….........................***H. aculeata* sp. nov.****5** Sternum VIII with posterior projection stout, almost 1/3 as wide as sternum, abruptly curved dorsally near apex......................***H. yasuni* Silveira, Lima, and McHugh, 2022.****5′** Sternum VIII with posterior projection slender, 1/4 as wide as sternum, not curved dorsally near apex (Figure 7G)…………...…..............................***H. amazonensis* sp. nov.****6** Sternum VIII with posterior projection reaching the posterior margin of pygidium; pygidium with central 1/3 almost straight…………...…...............................……….........................……….........................……….........................………..................***H. albertinoi* Silveira, Lima, and McHugh, 2022.****6′** Sternum VIII with posterior projection not reaching the posterior margin of pygidium; pygidium with central 1/3 emarginate…….........................................................................................................................................................................................***H. mendesi* Silveira, Lima, and McHugh, 2022.**

## 4. Discussion

Our results supported placing four new species in *Haplocauda*, which significantly expands the known richness and distribution of this genus. The finding of unusual terminalia and genitalia (such as the deep subcleft transverse groove [C66:S1], previously known only in *Scissicauda*) in some of the new species warranted an updated diagnosis for *Haplocauda*. Our findings also bear implications for the biology and taxonomy of the “*Scissicauda* lineage” in Lucidotini fireflies, as discussed below.

In the “*Scissicauda* lineage”, the genitalia, sternum VIII, and pygidium are important structures for genus- and species-level delimitation and identification. Where these structures present high interspecific morphological variability, such as in *Scissicauda* and *Haplocauda*, their function as clamps on the female abdomen during mating is suspected [5,13,26]. Unlike in *Scissicauda*, these traits are more variable than aedeagal ones among *Haplocauda* species, suggesting that these secondary sexual characters evolved faster than the primary traits. Further studies are needed to clarify the function(s) of this hypothesized copulation clamp and whether interactions between species and/or between males and females (sexual selection) may have led to reproductive isolation among *Haplocauda* species.

Our phylogenetic analyses consistently supported a late origin for the hypothesized copulation clamp in *Haplocauda*, given the more basal position of species lacking the typical abdominal modifications. Importantly, this is also evidenced in features such as the presence of anterior angles of the pygidium with blunt anterior projections and the presence of median projection in the posterior margin of sternum IX are not found in *H. lata* **sp. nov.** and *H. antimary* **sp. nov. **(Figure 7), species that are recovered at the base of the *Haplocauda* clade (Figure 1 and Figure 2). Since these structures are absent in the taxa recovered in the present analysis as basal to *Haplocauda*, it is possible that the ancestor of *Haplocauda* had relatively simpler terminalia and that these derived characteristics were fundamental for the divergence of the species.

*Haplocauda lata* **sp. nov.** was recovered at the base of the *Haplocauda* clade due to its simpler terminalia (as discussed above) and the combination of many atypical features of the genus, such as sternum VIII as long as Sternum VII (C27:S0), sternum VIII, with anterior 1/3 of pygidium distinctly narrower (C33:S1), syntergite with 1/2 of length to sternum IX (C47:S1), sternum IX with posterior margin emarginate (C54:S2), phallobase with sagittal line present (C57:S1), phallobase with apical margin deeply emarginate (C59:S1), dorsal plate of phallus with subcleft transverse groove deep (C66:S1). These traits are deemed plesiomorphic. Thus, it is possible that the ancestor of these two genera presented the dorsal plate of phallus with deep subcleft transverse groove that was maintained in *Scissicauda* and *H. lata* **sp. nov.** but has become shallower in the other *Haplocauda* species. The functional implications of this transformation remain unknown, but it is possibly related to sperm transfer since the ejaculatory duct in fireflies is usually sustained by the dorsal plate of the phallus.

The geographic ranges of species in the close relative taxa *Scissicauda* and *Haplocauda* overlap sometimes at some sites in the Amazon basin. Assuming syntopy (i.e., spatial and temporal co-occurrence), it is possible that reproductive isolation among species of these closely related lineages may be enabled by the unique species-specific morphology of the terminalia, which would prevent the formation of hybrids by mating incompatibility. Within *Haplocauda*, sympatry is seen between *H. albertinoi*, *H. amazonensis* sp. nov., and *H. lata* sp. nov., at the Adolpho Ducke Forest Reserve (Brazil: Amazonas), and *H. albertinoi* and *H. antimary* sp. nov., at the Antimary State Forest (Brazil, Acre). This raises interesting questions about the mechanisms of reproductive isolation: Are the differences in terminalia morphology sufficient to prevent hybrid formation between syntopic species? Or are there other prezygotic isolating mechanisms (e.g., pheromones [since these species do not have lanterns] or distinctive micro-habitat preferences) in addition to unique reproductive morphologies?

On the other hand, the fact that *H. yasuni*, *H. antimary* **sp. nov.** and *H. aculeata* **sp. nov.** are currently known only from their type locality—in sharp contrast to the more widespread *H. albertinoi* and *H. mendesi*—raises suspicions about allopatry, i.e., speciation due to geographic isolation between populations [27,28]. However, although allopatry plays an important role in the diversification and specialization of the morphological [29], behavioral and ecological [30] traits of Lampyridae, a crucial point to be investigated is the under-sampling artifact arising from the scarce amount of field data and targeted collection efforts.

Our study contributes to mitigating the pervasive Linnean shortfalls surrounding the lampyrids and demonstrates the critical relevance of the Amazonian region for the understanding of the evolution of fireflies.

## 5. Conclusions

Here, we expanded and adjusted the definition and diagnosis of *Haplocauda* after adding four new species, informed by a thorough and comprehensive phylogenetic analysis of morphological traits. Our results confirm the widespread occurrence of *Haplocauda* in the Amazon region and shed light over previously neglected patterns of co-occurrence in this genus.

## Figures and Tables

**Figure 1 insects-16-00824-f001:**
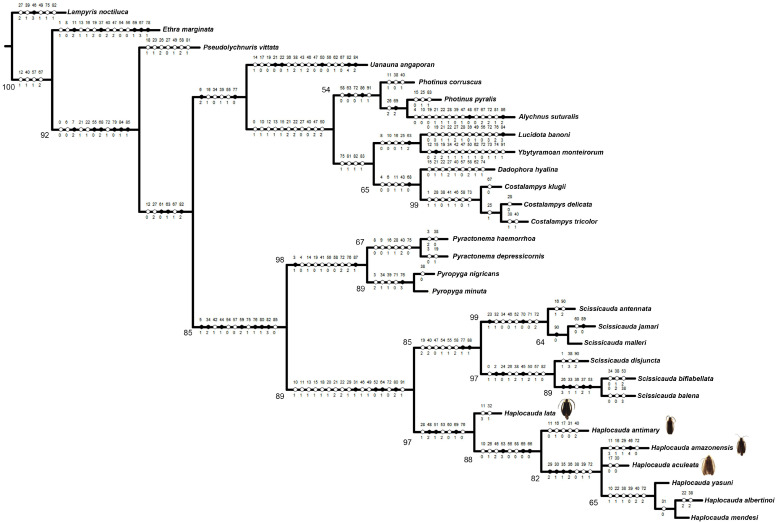
Phylogenetic relationship of *Haplocauda* using the implied weights Maximum Parsimony analysis (K = 4.765625). Synapomorphies that are not homoplastic are marked with black circles, while homoplastic ones are marked with empty circles. Below the branches are marked the symmetric resampling values.

**Figure 2 insects-16-00824-f002:**
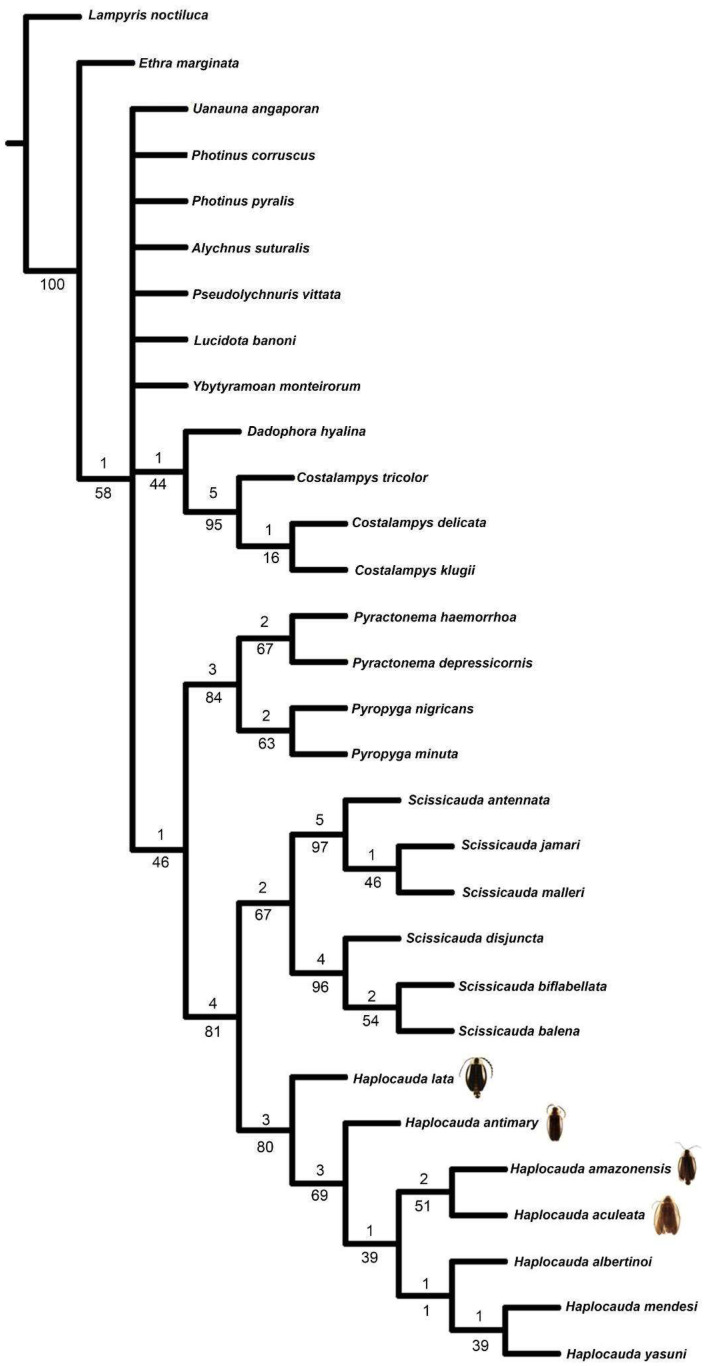
Consensus of the four most parsimonious trees found on the equal weights Maximum Parsimony analysis. The four new species described here are indicated by their dorsal habitus at their respective tips. Below branches, the absolute Bremer.

**Figure 3 insects-16-00824-f003:**
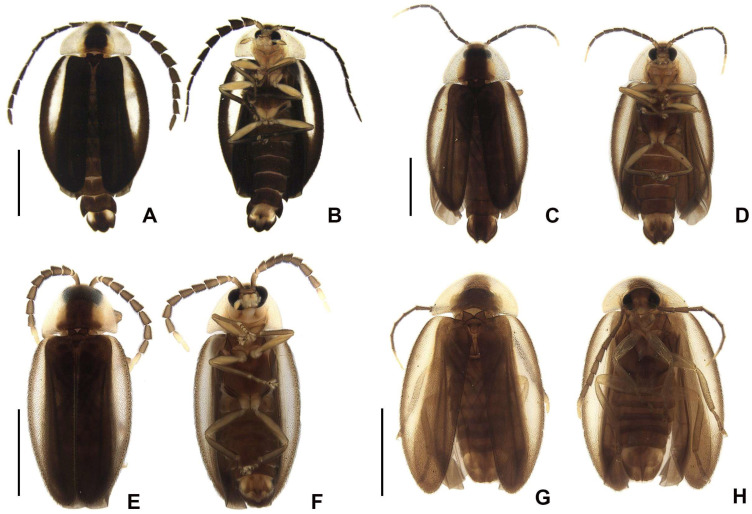
Habitus of *Haplocauda* spp, male. *H. lata* **sp. nov.**: (**A**) dorsal; (**B**) ventral; *H. amazonensis* **sp. nov.**; (**C**) dorsal; (**D**) ventral. *H. antimary* **sp. nov.**; (**E**) dorsal; (**F**) ventral; *H. aculeata* **sp. nov.**; (**G**) dorsal; (**H**) ventral. Scale bar: (**A,B**) = 3 mm; (**C**–**H**) = 2 mm.

**Figure 4 insects-16-00824-f004:**
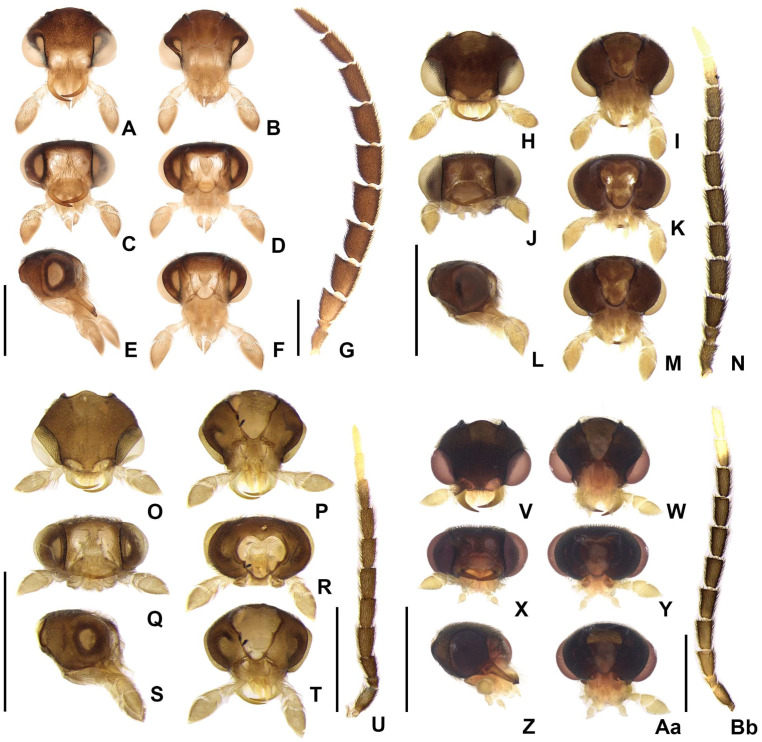
Diversity of head morphologies in *Haplocauda* spp. (**A**–**G**) *H. lata* sp. nov.; (**H**–**N**) *H*. *amazonensis* sp. nov.; (**O**–**U**) *H. aculeata* sp. nov.; (**V**–**Bb**) *H. antimary* sp. nov. Head, dorsal; (**A**,**H**,**O**,**V**), ventral; (**B**,**I**,**P**,**W**), frontal; (**C**,**J**,**Q**,**X**), posterior; (**D**,**K**,**R**,**Y**), lateral; (**E**,**L**,**S**,**Z**), occipital; (**F**,**M**,**T**,**Aa**) views. Left antenna, frontal; (**G**,**N**,**U**,**Bb**) view. Scale bars: (**A**–**F**,**G**,**H**–**N**,**O**–**T**,**U**,**V**–**Aa**,**Bb**): 1 mm.

**Figure 5 insects-16-00824-f005:**
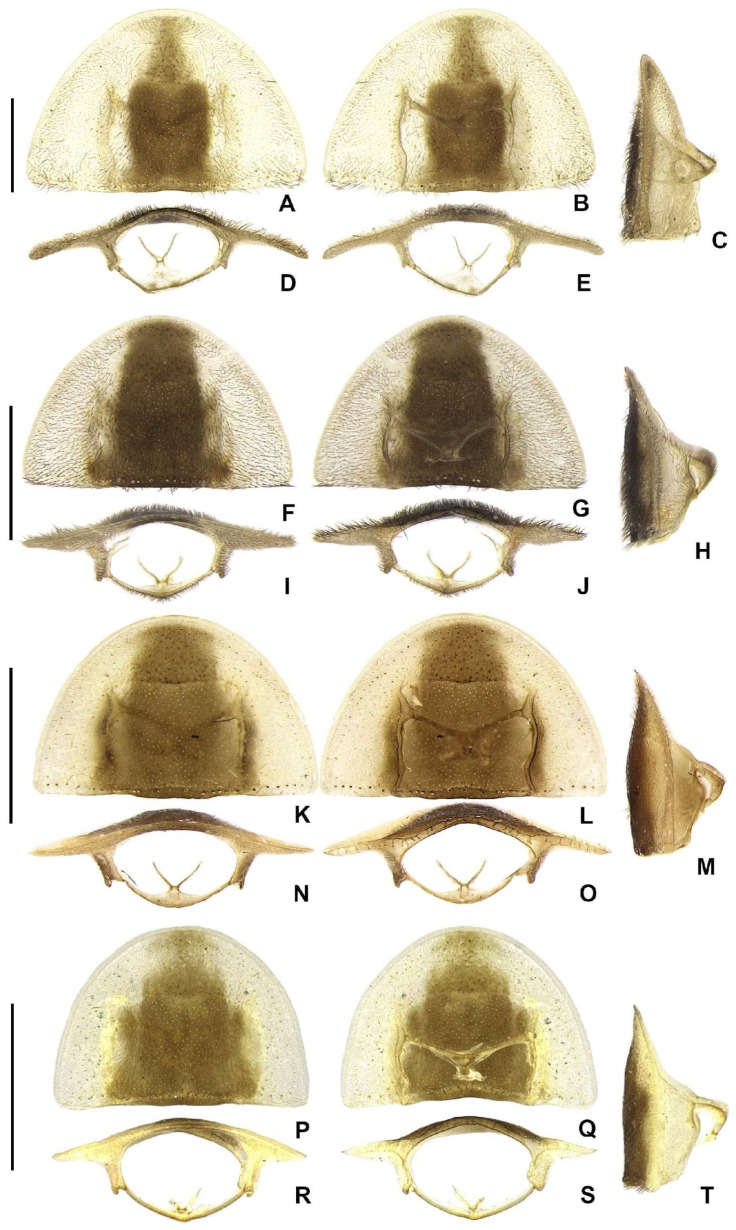
Diversity of pronotum morphologies in *Haplocauda* spp. (**A**–**G**) *H. lata* **sp. nov.**; (**F**–**J**) *H. amazonensis* **sp. nov.**; (**K**–**O**) *H. aculeata* **sp. nov.**; (**P**–**S**) *H. antimary* **sp. nov.** Pronotum, dorsal; (**A**,**F**,**K**,**P**), ventral; (**B**,**G**,**L**,**Q**), lateral; (**C**,**H**,**M**,**T**), anterior; (**D**,**I**,**N**,**R**), posterior; (**E**,**J**,**O**,**S**) views. Scale bars: (**A**–**E**,**F**–**J**,**K**–**O**,**P**–**S**): 1 mm.

**Figure 6 insects-16-00824-f006:**
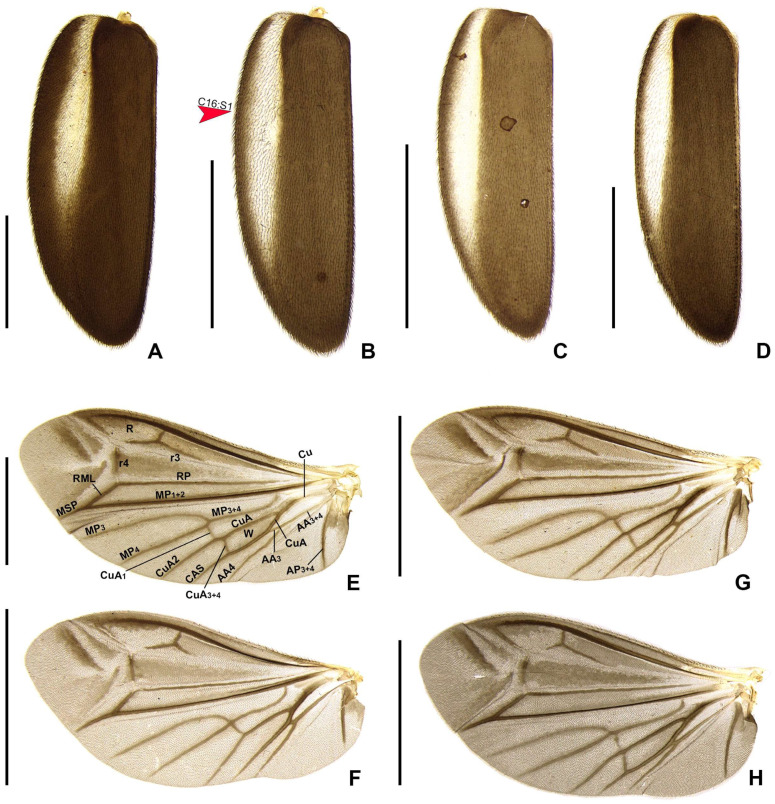
Diversity of elytron and wing in *Haplocauda* spp. (**A**,**E**) *H. lata* **sp. nov.**; (**B**,**G**) *H. amazonensis* **sp. nov.**; (**C**,**F**) *H. aculeata* **sp. nov.**; (**D**,**H**) *H. antimary* **sp. nov.**; (**A**–**D**) Left elytra, dorsal view; (**E**–**H**) Left wings, dorsal view. Scale bars: 2 mm.

**Figure 7 insects-16-00824-f007:**
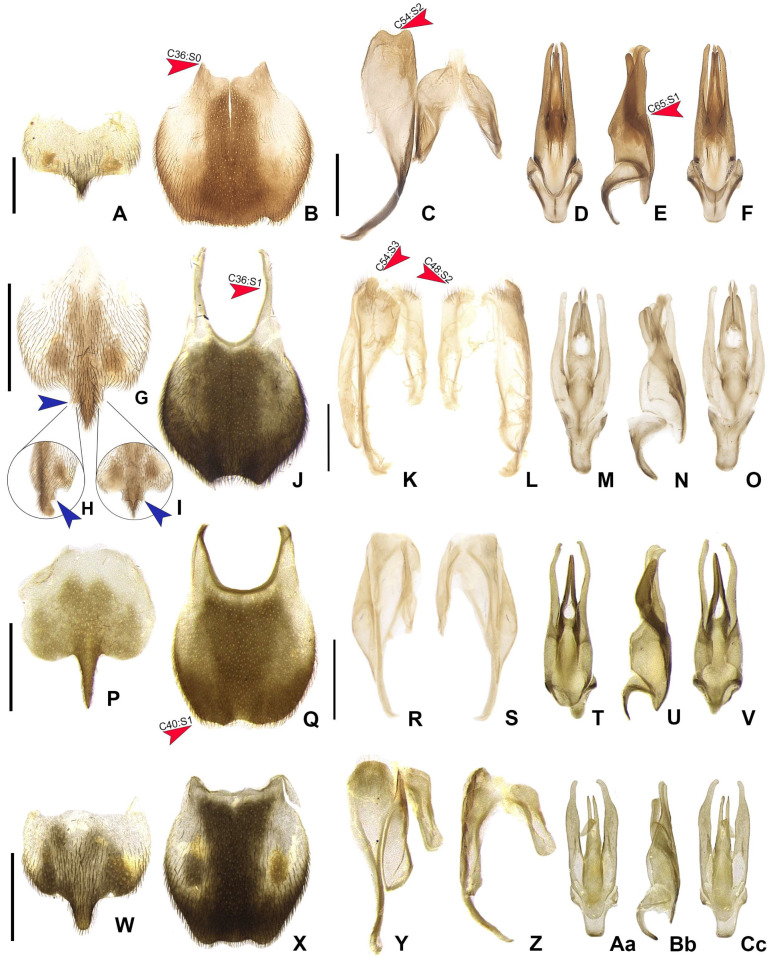
Diversity of terminalia in *Haplocauda* spp. (**A**–**F**) *H. lata* **sp. nov.**; (**G**–**O**) *H. amazonensis* **sp. nov.**; (**P**–**V**) *H. aculeata* **sp. nov.** (**W**–**Cc**) *H. antimary* **sp. nov.** Sternum VIII, ventral view; (**A**,**G**,**P**,**W**), ventrolateral view; (**I**) and dorsal view; (**H**). Pygidium, dorsal view; (**B**,**J**,**Q**,**X**). Syntergite, dorsal view; (**C**,**L**,**S**,**Z**), sternum IX, ventral view; (**C**,**K**,**R**,**Y**). Aedeagus dorsal; (**D**,**M**,**T**,**Aa**), lateral; (**E**,**N**,**U**,**Bb**) and ventral (**F**,**O**,**V**,**Cc**). Blue arrows showing the apophyses of the median projection of Sternum VIII. Scale bars: (**A**–**T**,**W**–**Cc**): 0.5 mm.

**Figure 8 insects-16-00824-f008:**
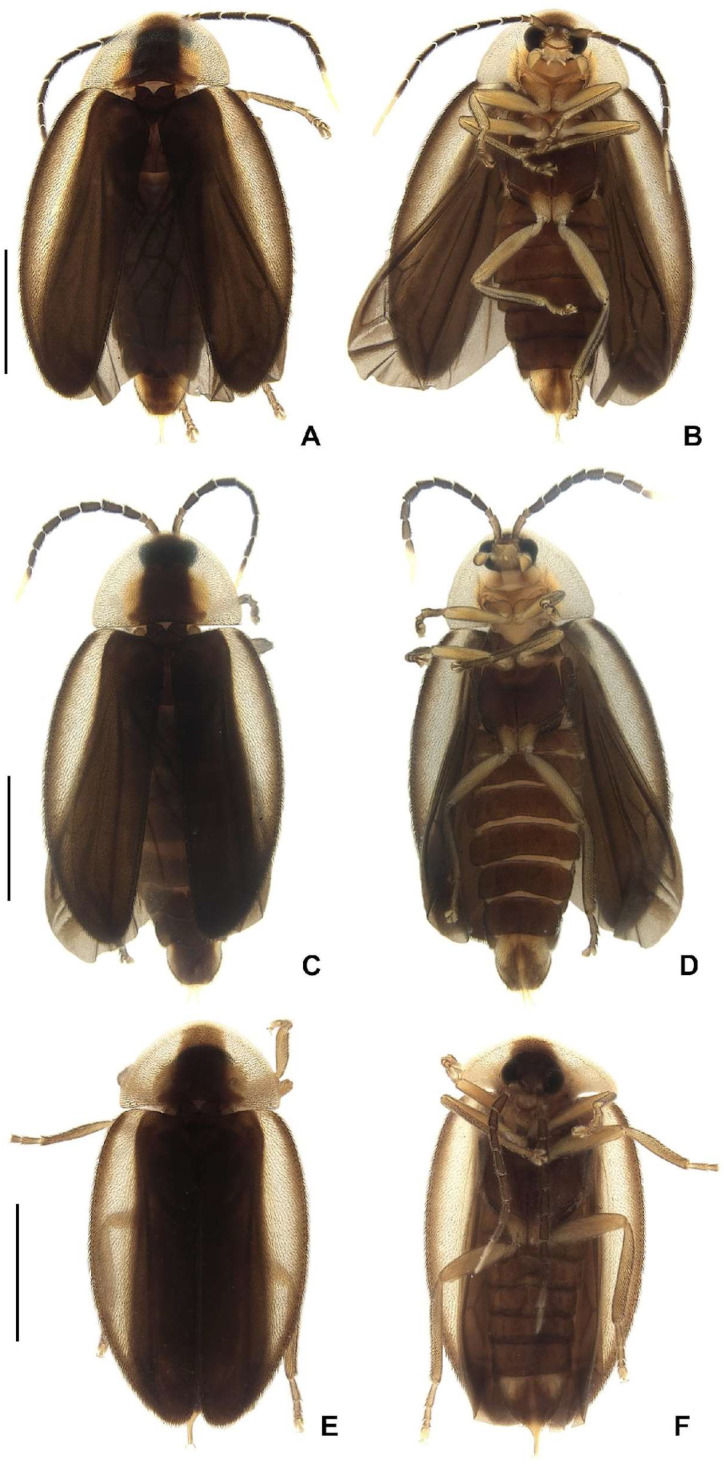
Habitus of *Haplocauda* spp, female. *H. lata* **sp. nov.**: (**A**) dorsal; (**B**) ventral; *H. amazonensis* **sp. nov.**; (**C**) dorsal; (**D**) ventral. *H. aculeata* **sp. nov.**; (**E**) dorsal; (**F**) ventral. Scale bar: (**A**–**F**)**:** 2 mm.

**Figure 9 insects-16-00824-f009:**
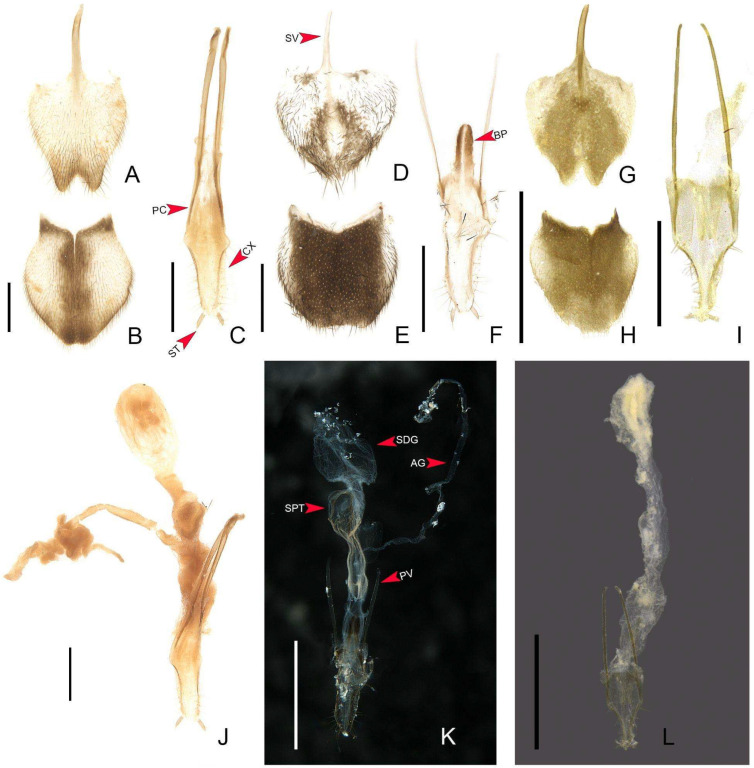
Diversity of terminalia in *Haplocauda* spp. (**A**–**C**,**J**) *H. lata* **sp. nov.** (**D**–**F**,**K**) *H. amazonensis* **sp. nov.** (**G**–**I**,**L**) *H. aculeata* **sp. nov.** Sternum VIII, ventral view (**A**,**D**,**G**). Pygidium, dorsal view (**B**,**E**,**H**). Genitalia, ventral view (**C**,**F**,**I**–**L**). Legend: AG: accessory gland; BP: bursa plates; CX: coxite; PC: paraproct core; PV: paraproctal valve; SV: spiculum ventral; SDG: spermatophore digesting gland; SPT: spermatheca; ST: stylus. Scale bars: (**A**–**J**,**L**): 0.5 mm; (**K**) 1 mm.

**Figure 10 insects-16-00824-f010:**
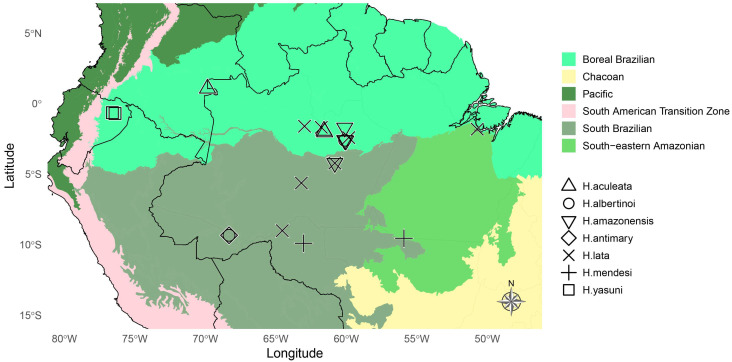
Geographic distribution map of *Haplocauda* spp. (including the four new species). Species of *Haplocauda* are more diverse in the Amazonian.

## Data Availability

The original contributions presented in this study are included in the article/Appendix A. Further inquiries can be directed to the corresponding author.

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
