# Peer review of "Four New Species of Haplocauda, with Notes on the Evolutionary Convergence of Copulation Clamps in Lucidotini (Coleoptera: Lampyridae: Lampyrinae)"

_insects, 2025, doi:10.3390/insects16080824_

Round 1

Reviewer 1 Report

Comments and Suggestions for Authors

Comments to the Authors

This manuscript describes four new species of the genus Haplocauda from the Amazon Basin and explores their phylogenetic placement, along with the relationships between Haplocauda and Scissicauda, based on a morphological phylogenetic analysis. An identification key to Haplocauda is also provided. The study represents a valuable contribution to our understanding of firefly diversity and evolutionary history in the Amazon region.

Some grammatical and formatting errors have already been annotated in the manuscript. Additionally, the authors should carefully verify and revise the following points:

Line 112: The authors mention the use of 95 morphological characters derived from two references, but Line 113 refers to the addition of four characters, with a final total of 92. This problem need author to check carefully.

Line 117: The authors state that TNT software was used for phylogenetic analysis. However, the specific version is not mentioned. Please clarify which version was used. Note that the most recent version is TNT v.1.6 (Goloboff & Morales, 2023; DOI: 10.1111/cla.12524), and the reference should be updated accordingly if applicable.

Figure 1: Please provide a clearer version of this figure.

Figure 10: Species names in the figure should be italicized and identified, e.g., H. lata sp. nov.

In addition, the following opinions need to be confirmed and discussed with the authors:

  1. It seems not very standard to incorporate the reference number into the text, for example, the yellow reference numbers are in lines 98 and 101.
  2. It seems like an extra word“anterior”, in the line 769, marked in blue, of the Key to Haplocauda Species Based on Males (Modified from Silveira et al., 2022).
  3. As shown in Figure 3, the four new male species differ greatly in the shape and color of their antennae, and the authors do not seem to have given a reasonable explanation about it.
  4. As shown in Figure 8, the paper shows only three female specimens of the new species, and the third specimen is missing its antennae, which the authors do not seem to explain.
  5. I don't seem to see from the paperhow many specimens were used in the experiment, especially the respective number of males and females.
  6. In the Remarkssection, the author briefly expounded the classification characteristics of the new species, but did not compare the classification characteristics with similar species, which made the establishment of the new species confused to some extent.
  7. At the same time, the author did not do molecular sequencing test for the four new species, which weakened the certainty of the establishment of the four new species
  8. In addition, the paper is too long, with a lot of text describing the genus of HaplocaudaSilveira, Lima and McHugh, 2022, from Line 335 to 480, and although the author fills in some new taxonomic information, it is questionable whether so much ink is necessary.
  9. As an important part of this paper, the discussion of evolutionary convergence of copulation clamps in Lucidotini,is only placed in the fourth part of Discussion, but not in the third part of Is that so? If so, I think it is inappropriate.

Reviewer 2 Report

Comments and Suggestions for Authors

The text is very well structured and written
My suggestions are:
a) delete the diagnosis, because it is not about description of Haplocauda, which was already done in 2022
The characters of the new species can be added in the redescription
b) suggest separating male and female characters in the character list
c) on the increase of sternum VIII (males), it is not clear whether it is an increase in width or length
d) the increase of the sternum VIII is in relation to the IX (it seemed to me), it is not very clear throughout the text
e) the captions of the figures need to be revised

Reviewer 3 Report

Comments and Suggestions for Authors

The MS presented has an overall merit acceptable, but the following points must be addressed: 
1. Reconsider the use of the simple summary; the Abstract has strong evidence of research, which is redundant
2. In M&M explicitly mention what the taxonomic terminology and criteria used, it isn't clear.
3. Species were all dry-pinned, or were some collected by the authors? make a statement covering this issue.
4. Eliminate diagnostic, just leave Description for each spp.

Comments on the Quality of English Language

Take care in M&M there are some finger errors

Round 2

Reviewer 1 Report

Comments and Suggestions for Authors

I'm basically satisfied with all the author's responses